# Antioxidant and Anti-Inflammaging Ability of Prune (*Prunus Spinosa* L.) Extract Result in Improved Wound Healing Efficacy

**DOI:** 10.3390/antiox10030374

**Published:** 2021-03-02

**Authors:** Sofia Coppari, Mariastella Colomba, Daniele Fraternale, Vanessa Brinkmann, Margherita Romeo, Marco Bruno Luigi Rocchi, Barbara Di Giacomo, Michele Mari, Loretta Guidi, Seeram Ramakrishna, Natascia Ventura, Maria Cristina Albertini

**Affiliations:** 1Department of Biomolecular Sciences, University of Urbino Carlo Bo, 61029 Urbino, Italy; s.coppari3@campus.uniurb.it (S.C.); mariastella.colomba@uniurb.it (M.C.); daniele.fraternale@uniurb.it (D.F.); marco.rocchi@uniurb.it (M.B.L.R.); barbara.digiacomo@uniurb.it (B.D.G.); michele.mari@uniurb.it (M.M.); loretta.guidi@uniurb.it (L.G.); 2Medical Faculty, Institute of Clinical Chemistry and Laboratory Diagnostic, Heinrich Heine University and the IUF- Leibniz Research Institute for Environmental Medicine Auf’m Hennekamp 50, 40225 Düsseldorf, Germany; Vanessa.Brinkmann@uni-duesseldorf.de (V.B.); Margherita.Romeo@iuf-duesseldorf.de (M.R.); natascia.ventura@uni-duesseldorf.de (N.V.); 3Center for Nanofibers and Nanotechnology, National University of Singapore, Singapore 119077, Singapore; seeram@nus.edu.sg

**Keywords:** polyphenols, MicroRNA, HUVEC, lifespan, *C. elegans*, tissue regeneration, cell migration, aging phenotype, biological aging

## Abstract

*Prunus spinosa L.* fruit (PSF) ethanol extract, showing a peculiar content of biologically active molecules (polyphenols), was investigated for its wound healing capacity, a typical feature that declines during aging and is negatively affected by the persistence of inflammation and oxidative stress. To this aim, first, PSF anti-inflammatory properties were tested on young and senescent LPS-treated human umbilical vein endothelial cells (HUVECs). As a result, PSF treatment increased miR-146a and decreased IRAK-1 and IL-6 expression levels. In addition, the PSF antioxidant effect was validated in vitro with DPPH assay and confirmed by in vivo treatments in *C. elegans*. Our findings showed beneficial effects on worms’ lifespan and healthspan with positive outcomes on longevity markers (i.e., miR-124 upregulation and miR-39 downregulation) as well. The PSF effect on wound healing was tested using the same cells and experimental conditions employed to investigate PSF antioxidant and anti-inflammaging ability. PSF treatment resulted in a significant improvement of wound healing closure (ca. 70%), through cell migration, both in young and older cells, associated to a downregulation of inflammation markers. In conclusion, PSF extract antioxidant and anti-inflammaging abilities result in improved wound healing capacity, thus suggesting that PSF might be helpful to improve the quality of life for its beneficial health effects.

## 1. Introduction

The wound healing process consists of different phases including hemostasis, inflammation, tissue proliferation, and tissue remodeling. Tissue injury activates the acute inflammatory response, which is necessary to provide tissue remodeling for repair. Successful repair requires the production of anti-inflammatory cytokines and downregulation of proinflammatory mediators. An excessive or prolonged inflammatory phase result in increased tissue injury and poor healing. The process depends on the persistence of inflammatory and oxidative stress conditions, which increase reactive oxygen species (ROS) formation leading to direct damage of cells [1].

Inflammaging, meaning that the aging process shows a chronic progressive proinflammatory phenotype [2], plays an increasingly important role in age-related diseases. In particular, inflammaging along with oxidative stress are among the main causes of chronic wound healing delay. Research in this area has attracted attention of academics in different fields of study revealing that several physiological and pathological mechanisms are supported by small non-coding RNAs, particularly microRNAs (miRNAs, post-transcriptional regulators of gene expression), which regulate inflammaging, as they are associated with many physiological processes correlated with aging, including cellular senescence. It has been previously demonstrated that miR-146a is associated with the human umbilical vein endothelial cell (HUVEC) senescent phenotype since it is highly downregulated in senescent cells [3]. Moreover, miR-146a is deeply involved in inflammatory responses as a very important actor in modulating the inflammation cascade since it can negatively regulate IRAK-1 expression [4].

Aiming at detecting novel agents to treat inflammatory diseases, several natural compounds or herbs have been tested to evaluate their anti-inflammatory potential. One of this is *Prunus spinosa* L. (also known as blackthorn, order Rosales, Rosaceae family, genus *Prunus,* species *P. spinosa*), a thorny shrub growing wild in uncultivated areas of Europe, West Asia, and the Mediterranean. The plant, used in phytotherapy for the treatment of cough but also as a diuretic, laxative, antispasmodic, and anti-inflammatory has, since a few decades, notably aroused the interest of the scientific community for its potential biological properties, and has been subject to extensive investigations [5,6,7,8,9,10]. For example, some of these authors have demonstrated a powerful antioxidant activity of *P. spinosa* fruit juice or extract (likely due to the presence of high levels of polyphenolic compounds) by DPPH (2,2-diphenyl-1-picrylhydrazyl) test. *P. spinosa* fruit (PSF) extract was also analyzed for bioactive effects in vivo on *Trichoplax adhaerens* (Placozoa) cultures and for antioxidant, antimicrobial, and anti-inflammatory activities in vitro with the aim to promote its adoption as new food beneficial for consumers or as a supplementary source of additive elements (i.e., natural pigments or antioxidants) for food or pharmaceutical industries.

In this study, our attention focused on a further characterization of PSF ethanol extract from our country (the Marche, Central Italy) in order to test its wound healing efficacy, after evaluating its antioxidant, anti-inflammatory, and antiaging properties, which are necessary for a good and effective wound healing activity. Wound healing is a primary therapeutic target for regenerative interventions, which aim to facilitate restoration of normal tissue architecture and function. Wound repair is a highly complex, multifaceted response, which is temporally and spatially coordinated across multiple stages through cooperative actions of diverse cell types, extracellular matrix molecules, cytokines, and growth factors [11]. Therefore, given increasing interest in identifying biologically active plant constituents for wound care, screening PSF ethanol extract for wound healing seemed particularly interesting and potentially a prelude to new and future applications.

The work hypothesis was that *Prunus* extract could exert a positive effect on wound healing as its bioactive polyphenols exhibit antioxidant, anti-inflammatory, and antimicrobial properties [12,13], which makes the extract ideal to play such a role in tissue repair.

Hence, after assessing the chemical composition of the extract, the study was organized considering either in vitro and in vivo experimental conditions in order to verify the beneficial properties of PSF, antioxidant (in vitro and in vivo), anti-inflammatory (in vitro), and antiaging (in vitro and in vivo) capacities. All these aspects were necessary to proceed the in vitro analysis of the PSF wound healing efficacy. Furthermore, in order to be able to correlate the wound healing activity with the anti-inflammaging property, after assessing the wound closure, the exact same PSF-treated cells were detached and used to evaluate the expression levels of markers associated to inflammation.

## 2. Materials and Methods

### 2.1. Chemicals and Reagents

Chromatographic solvents (water, acetonitrile, and formic acid) were HPLC grade, purchased from Merck (MERCK S.P.A., Milano, Italy).

### 2.2. Plant Material and Extract Preparation

*P. spinosa* L. ripe fruits were collected in November 2017 at “La Caputa” Urbania (PU, Marche, Italy), GPS coordinates: N43°40′46.512′′ E12°31′42.291′′ (350 m above sea level) and immediately frozen and stored at −20 °C until use. Voucher samples were deposited in the herbarium of Botanical Garden of Urbino University Carlo Bo: code “Pp125”. As for the extraction preparation, we followed the protocol previously described [13], except for one modification. In particular, we employed an acidified aqueous ethanol homogenizing solution (70 mL ethanol/29.9 mL distilled H_2_O/0.1 mL HCl 36%). In the end, the total amount of dried PSF extract obtained from 50 g of fruit and peel was 11.6 g.

### 2.3. Determination of Phenolic Compounds in Prunus Extract by HPLC/MS Analysis

The identification of phenolic compounds was achieved by comparison of the retention times and the spectra characteristics with those in data library [14] phenol express, and/or in literature, following a previously reported method [13]. Briefly, 10 mg of dried *Prunus* extract were dissolved in 1 mL of 70:30 *v/v* EtOH/H_2_O solution, and analyzed by HPLC-PDA/ESI-MS. The volume for injection was set at 50 μL. Before (direct) injection, all samples were cleared through a 0.45 μm nylon filter. Chromatographic analyses were performed using a MerkPurospher Star RP-18 endcapped column (4.0 mm × 250 mm, 5 μm particle size), thermostated at 25 °C, setting the elution flow rate at 0.7 mL/min. A system of two mobile phases was used for the elution: (A) aqueous 0.1% formic acid in double distilled water and (B) acetonitrile. The elution profile was as follows: 0–10 min, 94% A; 10–15 min, 83% A; 15–25 min, 78% A; 25–35 min, 76%A; 35–40 min, 74%A; and 40–45 min, 68%A.

The HPLC Waters system utilized for the analysis was equipped with an automatic injector, Alliance HT2795 separation module, column heater, Waters 2996 Photo Diode Array (PDA) connected to the mass spectrometer, Waters micromass ZQ MS system, and single quadrupole equipped with an electrospray ionization (ESI) interface. The instrument was operated in positive (ES+) and negative (ES-) ion mode with a scan range *m/z* 200–700. Capillary voltage was set at 3.50–3.51 kV (ES+) and 2.75–3.51 kV (ES-), source temperature at 100 °C and desolvation temperature at 300 °C. The cone and desolvation nitrogen gas flows were 50 and 500 L/h, respectively. UV–Vis spectra were recorded from 220 to 800 nm, with a bandwidth of 1.2 nm.

### 2.4. Cell Culture and Treatments

Human umbilical vein endothelial cells (HUVECs) derived from multiple donors were purchased from Life Technologies Corporation (Gibco™ C01510C, 1 × 10^6^ cells). Cells were seeded at a density of 5000/cm^2^ in T25 flasks (Corning Costar, Sigma Aldrich, St. Louis, MO, USA). HUVECs were cultured in the growth medium of Gibco ™ Medium 200 (Life Technologies Corporation, Grand Island, NY). The medium required the addition of the Gibco™ Low Serum Growth Supplement Kit (LSGS Kit) (Life Technologies Corporation, Grand Island, NY). The final concentrations of the components in the medium are fetal bovine serum (2% *v/v*), hydrocortisone (1 μg/mL), growth factor human epidermal (10 ng/mL), basic fibroblast growth factor (3 ng/mL), and heparin (10 μg/mL). Cells were incubated at 37 °C and 5% CO_2_ and allowed to grow to confluence as a monolayer. When cells reached 80% confluence, they were passaged with Trypsin-EDTA Solution (Sigma-Aldrich, St. Louis, MO, USA). All the experiments were performed using cells obtained from passage 3 (P3) to passage 15 (P5). HUVECs were treated with *P. spinosa* fruit (PSF) ethanol extract during a 6 h LPS (1 µg/mL) stimulation, while for replicative senescence and (P2–P15) wound healing, as specified in figure legends. PSF extract has been used at different concentrations (20/40/80 µg/mL): during 6 h for LPS stimulated HUVECs; 48 h for wound healing repair (as indicated below) and during P2-P15 replicative senescence.

### 2.5. In Vitro Wound Healing Assay

HUVEC cells were seeded in T25 tissue culture flasks, the cell monolayer was subjected to a mechanical scratch wound, horizontal along the flask, using the tip of a sterile pipette. HUVECs were treated with 20/40/80 µg/mL of ethanolic *P. spinosa* extract for 48 h. The untreated cells were used as a control and the vehicle condition was added too (ETOH). The images were acquired from the same lesion area immediately after scratching (t0) and after 48 h using a phase contrast microscope (Olympus Ix51 10× objective). We used ImageJ software to evaluate the total migration area during wound closure. The percentage of wound closure was calculated using the following formula: [(Wound area t0 − Wound area t)/ Wound area t0] × 100.(1)

The experiments were conducted in triplicate.

### 2.6. HUVEC Replicative Senescence Analysis

Each time that cells reached confluence, they were passaged through trypsinization, counted (using a Neubauer chamber) and replated. Replicative senescence was induced by culturing cells up to the 15th passage (160 days). The sum of all PD (population doubling) changes was used to calculate the cumulative population doubling level (CPDL).

SA-β-Gal analysis has been performed using the senescence cells histochemical staining kit (Sigma-Aldrich, St. Louis, MO, USA) where senescent cells positive for SA-β-Gal activity (colored in blue) were counted as a percentage related to the total cell number.

Cells were considered young when SA-β-Gal < 5%, and senescent when SA-β-Gal > 50%. SA-β-Gal activity was assessed as described previously [15].

### 2.7. Quantitative Real Time PCR (RT-qPCR) of Mature MicroRNAs and mRNA

The total RNA isolation kit (NorgenBiotek, Thorold, ON, Canada) was used to isolate total RNA (including both microRNA and larger RNA species) from HUVEC cells, following the manufacturer’s recommended protocol; RNA was stored at −80 °C until use. Human miR-146a and human RNU44 (reference miRNA) expressions were quantified using the TaqMan MicroRNA assay (Applied Biosystems, Foster City, CA, USA), as previously described [4].

The same total RNA isolation kit (NorgenBiotek, Thorold, Canada) was used to isolate total RNA from *C. elegans*. Total RNA was extracted from 1000 worms/condition after growing them for 3 days (from embryo) on plates with or without *Prunus* extract supplementation to the nematode growth media (NGM). *C. elegans* miR-124, miR-39, and U6 (reference miRNA) expressions were quantified using the same TaqMan MicroRNA assay mentioned before (Applied Biosystems, Foster City, CA, USA).

Isolated RNA was used to synthesize cDNA using a reverse transcription kit (Applied Biosystems, Foster City, CA, USA) according to the manufacturer’s protocol. Quantitative polymerase chain reaction real time (RT-qPCR) was performed with the SYBR Green PCR master mix (Applied Biosystems, Foster City, CA, USA) on an ABI Prism 7500 Real Time PCR System (Applied Biosystems, Foster City, CA, USA). The primers (IL-6: Forward AGGGCTCTTCGGCAAATGTA and Reverse GAAGGAATGCCCATTAACAACAA; IRAK-1: Forward CAGACAGGGAAGGGAAACATTTT and Reverse CATGAAACCTGACTTGCTTCTGAA) we used were the ones designed by Angel-Morales et al. [16]. TATA binding protein (TBP; Forward TGCACAGGAGCCAAGAGTGAA and Reverse CACATCACAGCTCCCCACCA) was used as an endogenous control to determine relative mRNA expression. The pairs of forward and reverse primers were purchased from (Sigma-Aldrich, St. Louis, MO, USA).

Product specificity was examined by the dissociation curve analysis. Results were calculated using the 2^−ΔΔCt^ method and are expressed as the fold change related to untreated control (CTRL).

### 2.8. DPPH Radical Assay

The antioxidant capacity of *Prunus spinosa* extract was evaluated by DPPH (diphenylpicrylhydrazyl) radical-scavenging method as in Fraternale et al. [17]. Of the reaction solution 280 µL was mixed with 80 µL of 0.5 mM DPPH (Sigma-Aldrich) ethanol solution and 40 µL of *P. spinosa* ethanol extract at different concentrations (from 12.5 to 100 µg/mL). Absorbance (A) was measured at 517 nm using ethanol as a blank. Of 0.5 mM DPPH 80 µL added to 320 µL of ethanol was used as negative control. DPPH inhibition (I, expressed in %) was calculated to evaluate the antioxidant activity by the following equation:I (%) = [(A_0_ − As)/ A_0_] × 100(2)
where A_0_ corresponds to the negative control absorbance and As corresponds to the tested sample absorbance. The higher the inhibition the higher the free radical scavenging ability. All analyses were performed in triplicate. L-ascorbic acid (Sigma-Aldrich, St. Louis, MO, USA) was used as a standard control at a concentration of 0.040 mg/mL.

The equation obtained by linear regression analysis was used to calculate the EC50 (considering y = 50).

### 2.9. C. elegans Cultivation

Wildtype (N2) nematodes were kept at 20 °C on NGM plates supplemented with 0.01% of ampicillin and 0.0005% of tetracycline *E. coli* HT115 (L4440) that was used as a food source.

### 2.10. E. coli Strains and Growth

HT115 (L4440) was obtained from Ahringer *C. elegans* RNAi feeding library [18]. It was grown in LB medium supplemented with 0.01% of ampicillin and 0.0005% of tetracycline at 37 °C overnight.

### 2.11. Prunus Extract Treatment in C. elegans

The *Prunus* extract was dissolved in 50% EtOH in a concentration 50 times higher than the desired final concentration. The extract was then spread on the NGM plates seeded with *E. coli HT115* (L4440). Of the stock solution 140 µL was spread on 7 mL NGM plates, while 360 µL was spread on 18 mL NGM plates.

### 2.12. Life and Healthspan

For lifespan and healthspan assays, age-synchronous populations were obtained by egg-lay on NGM plates containing the desired concentrations of *Prunus* extract. The treatment with *Prunus* extract was continued for the entire lifespan. Starting from adulthood, the worms were transferred to fresh NGM plates daily and the numbers of dead, alive, and censored animals were scored. For the healthspan analysis animals not moving, moving, and censored were scored. After the fertile phase, worms were transferred and scored every alternate day. Animals with internal hatching, an exploded vulva, or which died desiccated on the wall were censored. Survival analysis of pooled populations was performed in OASIS 2 [19,20].

### 2.13. Pharyngeal Pumping Rate and Body Bends Assay

Age-synchronized N2 nematodes were obtained by a worm bleaching protocol [21]. Eggs were then left to hatch during the night with a slight rocking. The day after, L1 larvae were spotted onto fresh NGM plates seeded with HT115 (L4440) and containing 400 µg/mL of *Prunus* extract or 1% ethanol (EtOH).

For the behavioral tests, nematodes (L4 larval stage) were collected and washed twice with M9 buffer to eliminate bacteria [22,23]. Worms were then incubated 2 h with 1 mM H_2_O_2_ (Sigma-Aldrich, St. Louis, MO, USA) in M9 buffer, on orbital shaking in dark conditions (100 worms/100 μL). Control worms were incubated with M9 buffer (vehicle) alone (100 worms/100 μL). After incubation, worms were transferred onto fresh NGM plates seeded with HT115 (L4440) and containing 400 µg/mL of *Prunus* extract or 1% ethanol. The body bends and the pharyngeal pumping rate were scored 20 h later [23,24]. Locomotor defects have been scored by measuring the number of body bends in liquid over a 1-min interval (body bends/min).

### 2.14. Statistical Analysis

All experiments were performed in triplicate if not differently specified. Statistical significance of *C. elegans* in vivo treatments was evaluated using the log-rank test. The *p*-values were corrected for multiple comparisons using the Bonferroni method. For the behavioral test, data were analyzed using GraphPad Prism 8.0 software (GraphPad Software, San Diego, CA, USA) two-way analysis of variance, and Bonferroni’s post hoc test. DPPH results are reported as linear regression analysis. For all the other experiments, statistical significance was determined at *p* < 0.05/0.01 using the Student’s *t*-test.

## 3. Results

### 3.1. Chemical Characterization

Chemical characterization of *Prunus spinosa* ethanolic extract was improved with respect to that previously described for a similar *Prunus* extract [13]. In particular, as reported in Table 1, the most abundant compounds among phenolic components (detected by HPLC–PDA/ESI–MS analysis) are the following: one isomer of caffeoylquinic acid, cyanidine-3-*O*-rutinoside, and peonidine-3-*O*-rutinoside (both showing anthocyanins typical UV absorption bands around 520 and 280 nm). The presence of a variety of quercetine derivatives, with higher retention time than the previous compounds, was also detected. In particular, based on retention time, typical flavonol UV absorption bands (around 260 and 350 nm), positive and negative molecular peaks, quercetine base peak 303 as a positive ion fragment, and the phenol express database (or references reported in Table 1), we tentatively identified the following derivatives: quercetine-3-O-rutinoside, quercetine-3-*O*-esoside-*O*-pentoside; quercetine-3-*O*-glucoside or galactoside, two quercetine-3-*O*-pentosides and a quercetine-3-*O*-pentoside derivative, and quercetine-3-*O*-ramnoside. These findings are in part coherent with previously reported analysis [13]. Despite this similarity, we could not detect cyanidine-3-*O*-glucoside, indicated as the main anthocyanin derivative often found in wild *Prunus* species. This could be due to a lower concentration of the latter compound in this local cultivar, and/or to technical restriction of the HPLC Waters system utilized for the chromatographic separation.

**Table 1 antioxidants-10-00374-t001:** Characterization of the main phenolic compounds in samples of *Prunus spinosa* ethanolic extract by HPLC–PDA/ESI–MS in positive and/or negative mode.

Peak No	*t*_R_ (min)	λ _max_ (nm)	M^+^ or [M+Na]^+^ (*m/z*)	[M-H]^−^ or [M+Na]^−^ (*m/z*)	HPLC-ESI/MS^n^ *m/z* (% base Peak)	Tentative Assignment	Ref.
1	15.2	300sh, 325	355 ^a^	353		O-Caffeoylquinic acid isomer	[1]
2	18.5	281, 514	595		MS^2^[595]: 287 (100), 449 (65)	Cyanidin-3-O-rutinoside	[25,26,27]
3	19.4	282, 517	609		MS^2^[609]: 301 (50), 463 (40)	Peonidin-3-O-rutinoside	[25,26,27]
4	25.9	260, 352	611 ^a^	609	MS^2^[611]: 303 (100), 465 (30)	Rutin (quercetin-3-O-rutinoside)	[1]
5	26.5	258,347	597 ^a^	595	MS^2^[597]: 303 (100), 465 ^a^(50)	Quercetin 3-O-hexoside-O-pentoside	[1]
6	27.4	258, 350	465 ^a^	463	MS^2^[463]: 301 20) MS^2^[465]: 303 (100)	Quercetin glucoside or galactoside	[1]
7	29.0	275, 350	625 ^a^	623 ^a^	MS^2^[623]: 433 (100), MS^2^[625]: 303 (100), 435 ^a^ (20)	Quercetin xyloside or arabinoside derivative	[1]
8	29.8	260, 352	435 ^a^	433	MS^2^[435]: 303 (100)	Quercetin arabinoside or xyloside	[1]
9	30.6	278, 352		433	MS^2^[435]: 303 (100)	Quercetin pentoside	[28]
10	31.4	278, 352		447	MS^2^[449]: 303 (100)	Quercetin-3-O-rhamnoside	[1]

^a^ Sodium adduct.

### 3.2. In Vitro Analyses

#### 3.2.1. In Vitro Evaluation of Antioxidant Properties

The polyphenols identified are in line with the high phenolic content previously indicated and the high antioxidant activity demonstrated in vitro by the DPPH free radical assay [13]. DPPH is a well-known radical and a scavenger for other radicals. Generally, rate reduction of a chemical reaction upon addition of DPPH is used as an indicator of the radical nature of that reaction. DPPH radical has a deep violet color in solution, and it becomes colorless or pale yellow when neutralized. The greater DPPH inhibition (I %), the greater the antioxidant activity of the substance tested. Obtained results showed that the extract effectively acts as an antioxidant in a dose-dependent manner (Figure 1) with an EC50 corresponding to 64.2 µg/mL.

#### 3.2.2. In Vitro Anti-inflammatory Activity

Considering the important role of oxidative stress and inflammation in the aging and wound healing processes, we first evaluated the anti-inflammatory effect of PSF in a classical cellular model, widely employed for aging and wound healing assays. In particular, young (P3) and senescent (P15) LPS-treated HUVECs, assayed separately, were employed. Senescent cells were identified based on the expression of senescence-associated biomarkers, including the senescence-associated secretory phenotype (SASP) (SA-β-Gal > 50%), (data not shown). In P3 cells, during an acute (6 h treatment) proinflammatory stimulus, PSF ethanol extract at different concentrations (20 µg/mL; 40 µg/mL; 80 µg/mL) increased miR-146a and decreased IRAK-1 and IL-6 expression levels (Figure 2). Although the real nature of molecular interactions between TLR4 and *P. spinosa* ethanol extract are still to be investigated, our findings showed, as observed in U937 cells under the same experimental conditions [13], that in HUVEC cells as well, *Prunus* extract can upregulate intracellular miR-146a, with a consequent downregulation of the TLR-NF-κB-mediated inflammatory response, particularly by inhibiting TLR4 signaling pathway and reducing cytokine (IL-6) production. We performed the same treatment on older cells. As shown in Figure 2, LPS stimulated P15 HUVECs showed upregulated miR-146a levels (LPS) that even increased with PSF extract treatment (20/40/80 µg + LPS). In addition, obtained results in both young and senescent HUVEC cells revealed that PSF was able to decrease IRAK-1 and IL-6 expressions, thus strongly suggesting an anti-inflammatory activity, which makes PSF extract a good candidate for wound healing.

### 3.3. In Vivo Analyses

Taking into account that due to cost and duration, relatively little is known about whether dietary polyphenols are beneficial in whole animals, particularly with respect to aging, to test the potential bioeffects (i.e., antioxidant and antiaging activity) of PSF, in addition to the in vitro model, we also employed the nematode *Caenorhabditis elegans*, a powerful model organism widely used for aging and longevity studies, due to its short (2–3-week) lifespan, rapid generation time, and experimental flexibility [29,30]. In particular, *C. elegans* is an excellent and useful model employed to analyze and understand organismal responses to different natural and synthetic compounds, including polyphenols, and their influence on aging and lifespan, as different biological processes and numerous aspects of aging are very similar in nematodes and mammals, including humans [31]. Significant changes in gene expression occur during *C. elegans* aging. In fact, the distinct age-dependent gene expression profiles that exist between genetically identical individuals are likely to be mediated through variations in gene regulatory networks and, once again, microRNAs represent likely candidates for mediating some of these variations even by modifying *C. elegans* lifespan. For example, overexpression of miR-124 causes the lifespan extension of *C. elegans* [32], whereas the loss of miR-124 increases ROS formation and accumulation of the aging markers, resulting in a reduction in lifespan [33]. Furthermore, miR-39 has been shown to be involved in strong age-specific changes [34].

#### 3.3.1. Longevity Study

To strengthen the beneficial antiaging effects of PSF extract in vivo, we moved to a powerful model organism widely used for aging and longevity studies, the nematode *C. elegans*. Lifespan analysis was performed using wild-type worms exposed to different concentrations of the extract throughout the entire lifespan. When examining living organisms, the bioavailability has a key role while evaluating health effects. That is the reason why concentrations employed in in vivo assays are different and (often) higher that those tested in vitro. At the moment PSF bioavailability is unknown, therefore starting from a concentration (40 µg/mL) that had proved to be effective in vitro or in vivo experiments [12,13], we decided to test also a ten-fold lower (4 µg/mL) and a ten-fold higher one (400 µg/mL) on *C. elegans*.

Figure 3A shows the survival curves of treatment groups against the control group (ETOH) plotted by the Kaplan–Meier model. In particular, *C. elegans* treated with different concentrations (4, 40, and 400 µg/mL) of *Prunus* extract did not show any differences from each other and from CTRLs up to the first 10 days. While, worms treated with the highest concentration (400 µg/mL) showed a lifespan significantly extended (i.e., a right-shifted curve, Bonferroni *p*-value ETOH vs. 400 µg/mL 1.2 × 10^−7^) compared with the other experimental conditions.

The control group of nematodes without the *Prunus* extract had a mean lifespan of 20.26 ± 0.48 days under the proper survival conditions. Compared with the control, the highest concentration (400 µg/mL) group significantly extended the mean lifespan to 24.71 ± 0.48 days and the survival rate increased by 22%.

Moreover, healthspan, here defined as the time when the nematodes are still actively moving, was prolonged after treatment with PSF extract. As shown in Figure 3B, 400 µg/mL of *Prunus* extract significantly (Bonferroni *p*-value ETOH vs. 400 µg/mL 0.0007) increased the number of healthy worms with respect to the other experimental conditions. While the mean healthspan of control worms was 17.39 ± 0.41 days, worms treated with 400 µg/mL PSF had a healthspan of 20.18 ± 0.40 days, which is an increase of 16%.

#### 3.3.2. In Vivo Evaluation of Antioxidant Properties

Taking into account that the antiaging effect of a given compound is often associated with the ability to increase stress resistance, to validate the antioxidant effects of PSF in vivo, we tested *C. elegans* resistance to H_2_O_2_ induced oxidative stress. To this aim, four-day old (L4) larvae of wild type (N2) nematodes were treated for 2 h with 1 mM H_2_O_2_ or M9 buffer (vehicle) and then moved back onto NGM plates containing 400 µg/mL *Prunus* extract or 1% EtOH. The behavioral tests were performed 20 h later. Wild type worms treated with the extract at the highest concentration (400 µg/mL) exhibited an enhanced resistance to H_2_O_2_ induced oxidative stress in comparison with untreated worms. Specifically, the detrimental effects on vital parameters (i.e., pharyngeal pumpings/min and body bends/min), induced by 2 h of treatment with peroxide, were significantly prevented in individuals fed with the *Prunus* extract compared to control vehicle (ETOH) fed animal (Figure 4A,B).

#### 3.3.3. In Vivo MicroRNAs Modulation Associated to *Prunus* Antioxidant Activity

It has already been demonstrated that overexpression of miR-124 causes the lifespan extension of *C. elegans*, suggesting that miR-124 may be involved in regulation of longevity in *C. elegans* [32] and that the loss of miR-124 in *C. elegans* increases ROS formation and accumulation of the aging markers, resulting in a reduction in lifespan [33]. Furthermore, miR-39 has been shown to be involved in modulation of strong age-specific changes [34]. On this basis, we decided to analyze the expression of miR-124 and miR-39 in PSF treated *C. elegans*. As indicated in Figure 4C, miR-124 expression was increased after *P. spinosa* treatment, while miR-39 was decreased. Both results seem to suggest a potential positive effect of *Prunus* extract on *C. elegans* lifespan by the modulation of miR-124 and miR-39 expression levels. Hence, on the whole, our findings revealed that PSF extract may have potential antiaging and antioxidant effects in vivo through extending the lifespan, promoting the healthspan and, still, enhancing stress resistance of *C. elegans*.

### 3.4. PSF Extract Effect on Wound Healing

In order to better verify the correlation among PSF antioxidant, anti-inflammatory, and antiaging effects with wound healing capacity, once validated PSF beneficial effects either in vitro or in vivo, we moved to the principal goal of the study, which was to evaluate the activity of the extract on wound healing. In fact, the main hypothesis of the study was that a significant wound healing activity was to be expected based on PSF antioxidant and anti-inflammaging ability tested both in vitro and in vivo models, as herein previously reported. Noteworthy, our results showed that treatment with *P. spinosa* extract significantly improves wound healing closure both in P3 and P15 cells. In particular, concentrations of 40 and 80 µg/mL determined a significant increase in the percentage of wound healing compared to CTRL and ETOH groups. PSF treatment showed, albeit with some slight differences considering the concentrations used, a wound healing closure percentage around 70% both in younger and older cells, thus revealing an enhanced cell migration, which was unexpected in a relatively advanced phase of cellular aging (Figure 5A).

To better investigate a hypothetical mechanism of action, miR-146a along with IRAK-1 and IL-6 amounts, were analyzed in wound healing HUVECs as well. Young P3 cells treated with *Prunus* extract showed a significant increase in miR-146a expression level, with evident lowering of IL-6 and IRAK-1 amounts. Similar results were obtained treating senescent (P15) cells with PSF, indicating an anti-inflammatory activity during wound healing (Figure 5B–D). As concerns data illustrated in Figure 5B,C, showing that IL-6 and IRAK-1 were downregulated not only by PSF extract but also by EtOH alone (the vehicle), in our opinion this might be due to a technical bias (see discussion section).

### 3.5. HUVEC Replicative Senescence Analysis and PSF Effect on Cell Biological Aging

HUVECs were also maintained in long-term cultures until growth arrest (XV passage) to mimic cellular aging, with cultures being supplemented with PSF ethanol extract at different concentrations (20, 40, and 80 µg/mL) for the entire period (160 days). Population doubling and senescence-associated β-galactosidase (SA-β-gal) staining, indicated a progressive acquisition of senescence status. A long-term treatment with PSF extract for nearly 160 days did not significantly increase HUVEC proliferation capability as indicated by the cumulative population doubling level (CPDL) (Figure 6A). Moreover, the extract did not delay the onset of aging phenotype shown by senescence-associated β-galactosidase (SA-β-gal), thus suggesting that PSF treatment does not seem to affect and/or change the senescence process (Figure 6B). Noteworthy, however, *Prunus* extract appears to have an important “qualitative” effect on cell differentiation, at least at certain concentrations. In fact, while, on the one hand, older cells treated with the extract at 20 µg/mL did not differ from controls (and ETOH group) in morphology showing a typical cobblestone phenotype, interestingly, on the other hand, when treated with 40 or 80 µg/mL appeared as long cells with multiple protrusions connecting neighboring cell clusters (Figure 6C). These results lead us to think that the antioxidant and anti-inflammatory effects of PSF extract may allow older cells to behave as if they were younger, coping with the proinflammatory phenotype associated to cellular senescence. In fact, although the cells treated with the extract age at the same speed as the controls, they behave and act as if they were “biologically” younger.

When examining the cellular senescence at the molecular level, in line with previous experimental data [4], older cells showed a proinflammatory, pro-oxidative senescence-associated phenotype (SASP), i.e., a proinflammatory condition characterized by high IRAK-1 and IL-6 amounts (see Figure 7, CTRLs). Notably, after PSF extract treatment, the modulation of miR-146a and IL-6 and IRAK-1 was comparable to what was observed in younger cells. More in detail, considering the effect of *P. spinosa* on HUVEC senescence without any additional triggers, although due to the current situation (i.e., COVID-19 emergency) our data were absolutely preliminary (one single experiment), the extract seemed to act as an anti-inflammatory agent, as shown by IL-6, IRAK-1, and miR-146a expression levels, evaluated in P2, P5, and P15 cells (Figure 7).

## 4. Discussion

Cellular senescence is a persistent hyporeplicative state, first described by Hayflick [35], that can be induced by several types of internal or external (environmental) stress. At present, senescent cells are mostly identified by the combined presence of multiple traits, including senescence-associated protein expression and secretion, DNA damage and β-galactosidase activity. Senescent cells are known to adopt a secretory phenotype that comprises a large number of potent proinflammatory, angiogenic, and tissue-remodeling factors. This important trait, reported as the senescence-associated secretory phenotype (SASP), influences many biological processes such as tissue repair and regeneration, tumorigenesis, and the aging-associated proinflammatory state. At a biological level, senescence is a two-sided medal playing and important role in physiological and pathological processes: on the one hand it is beneficial for tissue remodeling, embryonic development, wound healing, and tumor suppression in young individuals [36,37,38], on the other hand, in old individuals, it promotes aging-associated declines and diseases including age-related chronic inflammation (i.e., neurodegeneration) [39,40] and cancer.

Oxidative stress and inflammation are the major mechanisms of endothelial dysfunction with advancing age and oxidative stress contributes to chronic inflammatory processes implicated in aging, referred to as “inflammaging”. Plant polyphenolics have been shown to modify signal transduction pathways contributing to delaying or preventing inflammation [13,41,42]. In this study, the potential antioxidant, anti-inflammatory, and antiaging effects of *P. spinosa* L. fruit (PSF) ethanol extract (particularly rich in polyphenolics) were investigated in vitro and in vivo. Our results demonstrated that PSF extract exerted antioxidant activities, along with favorable anti-inflammatory and antiaging properties.

In line with available data reporting that plant polyphenols have anti-inflammatory effects via NF-kB-dependent mechanisms (i.e., by downregulating the NF-kB inflammatory cascade), at a molecular level, PSF extract exerts protective (anti-inflammatory) effects on HUVEC cells through modulation (i.e., upregulation) of miR-146a expression levels, along with a consequent downregulation of IRAK-1 and inhibition of production of the proinflammatory cytokine IL-6.

Noteworthy, HUVEC cells senescence, besides high levels of β-galactosidase and low levels of telomerase expression, is associated with a SASP mainly characterized by high levels of IL-6 expression [4]. Moreover, aging-associated miRNAs are largely negative regulators of the immune innate response and target central nodes of aging-associated networks. In particular, miR-146a exerts its negative regulatory action on NF-κB, the downstream effector of TLR signals that leads to the induction of proinflammatory responses. While miR-146a expression levels in studies of cellular senescence are controversial [3,4] and interpreting such a controversy is beyond the scope of this study, it is very likely that high quantities of miR-146a reported in senescent cells might be explained by hypothesizing that miR-146a is also involved in alternative mechanisms.

*C. elegans* is a powerful model organism largely employed for aging studies. In this work, we identified PSF as a new polyphenolic compound with potent antioxidant and antiaging properties, two parameters that often strongly correlate in longevity studies. The in vivo beneficial effects of this dietary polyphenolic compound open the door for future studies aiming at addressing the impact on other important age-associated features and on the molecular mechanisms underlying its beneficial effects. To this extent, it is worth noting that besides their antioxidant activity, many substances, including polyphenols, exert their prolongevity effect through induction of autophagy, a key recycling process, which helps dampening damaged or old intracellular components [20,43]. While we have not tested this possibility yet, here we show that PSF may modify gene expression epigenetically by microRNAs modulation and, interestingly, miRNAs have been shown to modulate autophagy in different species, including *C. elegans* [44]. Specifically, we found that PSF modulated the expression of two important markers associated with aging and oxidative stress in *C. elegans,* i.e., miR-124 (upregulated) and miR-39 (downregulated), thus suggesting an interesting mode of action through which these dietary polyphenols could modulate aging and associated features [32,34].

Noteworthy, besides the antioxidant property, the activity of PSF polyphenolics in regulation of mechanisms that decrease the inflammation cascade resulted to promote wound healing (a complex molecular and cellular process made of several overlapping phases including inflammation) through cell migration, thus confirming our initial work hypothesis, which considered PSF as a good candidate for wound care and tissue regeneration. In particular, our results demonstrated that PSF extract enhanced endothelial wound closure (up to 70%) and cell migration, which are important properties of endothelial cells in order to be resilient to tissue injury [45]. Intriguingly, one unforeseen outcome was the high percentage of wound closure observed not only in young cells but also in older ones. Indeed, the results were almost overlapping in all cells, albeit with some small differences depending on the concentration of the extract. Moreover, in wound healing cells, expression levels of inflammation-associated biomarkers (miR-146a, IL-6, and IRAK-1) confirmed PSF anti-inflammaging ability, which, combined to its antioxidant capacity, is at the basis of the improved wound healing efficacy observed after PSF treatment. However, results observed in IL-6 and IRAK-1 expression levels may appear, in part, unexpected. In fact, as shown in the results section (Figure 5B-C), IL-6 and IRAK-1 were downregulated not only by PSF extract but also by EtOH alone (the vehicle). Given the proinflammatory effect of EtOH, one would have expected exactly the opposite (i.e., high levels of IL-6 and IRAK-1). However, considering that this apparent “anomaly” occurred only in PCR reactions performed on (messenger or micro) RNA extracted from HUVEC cells, a possible explanation is that it is a technical bias (i.e., in the case of real time PCR, ethanol acts as a reverse transcriptase inhibitor). This would be indirectly confirmed by the fact that no anomalies were noted in the experiments carried out directly on whole cells (treated with non-toxic EtOH concentrations). Finally, the fact that molecular amplifications on *C. elegans* did not show any bias could be due to the fact that in vivo the penetration of EtOH into the living tissues is considerably lower than that which occurs in an in vitro system, thus reducing the risk of PCR inhibition. On the whole, although at present data are still preliminary and further investigation on this item will be necessarily needed, our findings suggest that *Prunus* extract can be considered among the products that, at appropriate doses, could be useful to prevent endothelial dysfunction, which is one of the major contributors to the development and progression of atherosclerosis and other cardiovascular (CVD) or neurodegenerative (i.e., Alzheimer’s or Parkinson’s) diseases.

When examining senescence of HUVECs treated with *Prunus* extract at different concentrations during 160 days (P2-P15), contrary to other reports [46], PSF extract did not delay the onset of the aging phenotype shown by senescence-associated β-galactosidase (SA-β-gal), but, rather, appeared to have an important “qualitative” effect on cell differentiation. In fact, P15 cells grown in a culture medium supplemented with PSF at 40 or 80 µg/mL showed (compared to the controls) a different phenotype characterized by long cells with multiple protrusions connecting neighboring cell clusters, revealing an unexpected differentiation capacity. Although this finding was certainly stimulating, the great difficulty of attending the laboratory with diligence, owing to the closure of the university facilities due to the COVID-19 emergency, created considerable problems in carrying out the first experimental test (which took 160 days) and made it absolutely impossible to repeat the entire experiment in triplicate. Unfortunately, the current situation is still very unstable and therefore we cannot provide more in-depth data at the moment. However, even if we are aware that this data could be omitted, the result itself appears so interesting that it convinced us to communicate it anyway, albeit in a preliminary way.

If confirmed, this outcome could be interpreted as related to the positive effect of polyphenols on microvascularization by enhancing cell migration [45]. Moreover, such an effect could be miR146a–mediated. In fact, it has been demonstrated that miR-146a upregulation promotes angiogenesis by inducing FGFBP1/FGF2 chemokine signaling events that are involved in the promotion of HUVEC proliferation, tube formation, and migration [45]. Furthermore, we suggest that the antioxidant and anti-inflammatory effects of PSF extract may allow older cells to behave as if they were younger, coping with the proinflammatory phenotype associated to cellular senescence. Indeed, although the cells treated with the extract age at the same speed as the controls, they behave, act, and react as if they were (biologically) younger. Therefore, it is as if the extract, while not changing the chronological age of the cells, can somehow act on their biological age by setting the functional clock back, probably due to the anti-inflammatory effect. This idea is also corroborated by the results obtained in the analysis of miR-146a, IL-6, and IRAK-1 expression levels in older cells (P15), which are much less inflamed than controls (i.e., levels of miR-146a are significantly higher and those of IL-6 and IRAK-1 lower). Briefly, CTRL and PSF-treated cells having the same chronological age end up not having the same biological age.

Biological aging (i.e., that aging occurs as you gradually accumulate damage to various cells and tissues in the body) is quite different from chronological age (the amount of time that has passed from your birth to the given date). Additionally, known as physiological or functional age, biological age differs from chronological age because it takes into consideration a number of factors other than just the day you were born. Chronological age will always be an easy-to-determine number, while biological age depends on a number of variables that can change on a continuing basis. Environmental factors may affect aging, thus making biological aging and chronological aging quite distinct concepts. Although more studies need to be done on factors for determining biological age, it has been suggested that there is a clear link between nutrition and biological age; therefore, being actively aware of what constitutes a healthy diet may help to improve biological age [47]. Speaking of which, we believe that PSF extract might be suggested in a healthy diet for healthy consumers (Figure 8).

In our study, PSF extract (abundant in polyphenolics) acted as an antioxidant and anti-inflammatory agent in vitro and in vivo, showing also a beneficial effect on lifespan and healthspan of *C. elegans*; moreover, PSF treatment enhanced the cellular response to wound healing, suggesting a potential activity of polyphenols to promote endothelial cell functions. In this regard, it would be very interesting to verify in a future study the hypothesis that PSF treatment may result in a significant modification of capillary-like tube formation (angiogenesis) by inducing important molecular markers responsible for the microcirculation and (maybe) vasodilation of endothelial cells, thus showing meaningful benefits on microvascular activities. To test the ability of the PSF polyphenols to affect microcirculation and vasodilation in endothelial cells, many experiments need to be carried out including: gene expression analyses of the vascular endothelial growth factor (VEGF, well known for its role in angiogenesis and wound repair), the endothelin 1 (ET-1), the endothelial nitric oxide synthase (eNOS), and the miR-146a-CREB3L1-FGFBP1 signaling axis. In the event that such a conjecture is confirmed, it would be possible to affirm that a dietary supplementation with foods containing PSF extract, properly prepared, could be an inexpensive, non-pharmacological approach for improving cardiovascular health not only in currently healthy individuals but also in populations with microvascular dysfunction.

## 5. Conclusions

Present findings provide further support to the view that plant polyphenols may affect inflammation and associated disorders not only as antioxidants but also as modulators of inflammatory redox signaling pathways with a positive effect on wound healing. In particular, our results suggest that, at a proper dilution, *Prunus spinosa* may have health benefits and could be a potential dietary supplement and alternative medicine with antioxidant, antiaging, and wound healing properties to be included in those lifestyle interventions useful in the treatment of acute inflammation and senescence associated diseases and/or tissue repair and regeneration. However, although our results might be of interest to prevent and/or treat aging-associated disorders and might suggest a way to improve the quality of life, further in-depth mechanistic and translational studies with other organisms, including humans, are needed to support these beneficial effects and to determine the clinical relevance.

## Figures and Tables

**Figure 1 antioxidants-10-00374-f001:**
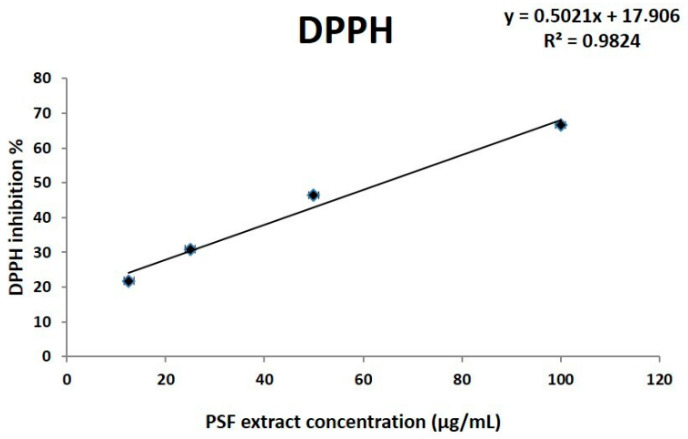
DPPH analysis of *P. spinosa* extract antioxidant activity was carried out at different *P. spinosa* extract (PSF) concentrations (from 12.5 to 100 µg/mL). DPPH inhibition (I, expressed in %) was used to evaluate PSF antioxidant activity. Linear regression analysis (R^2^ = 0.982) shows an excellent concentration-dependent effect. *P. spinosa* exerts an antioxidant activity with an EC50 = 64.2 µg/mL (calculated (given y = 50) as: 50 = 0.502x + 17.906).

**Figure 2 antioxidants-10-00374-f002:**
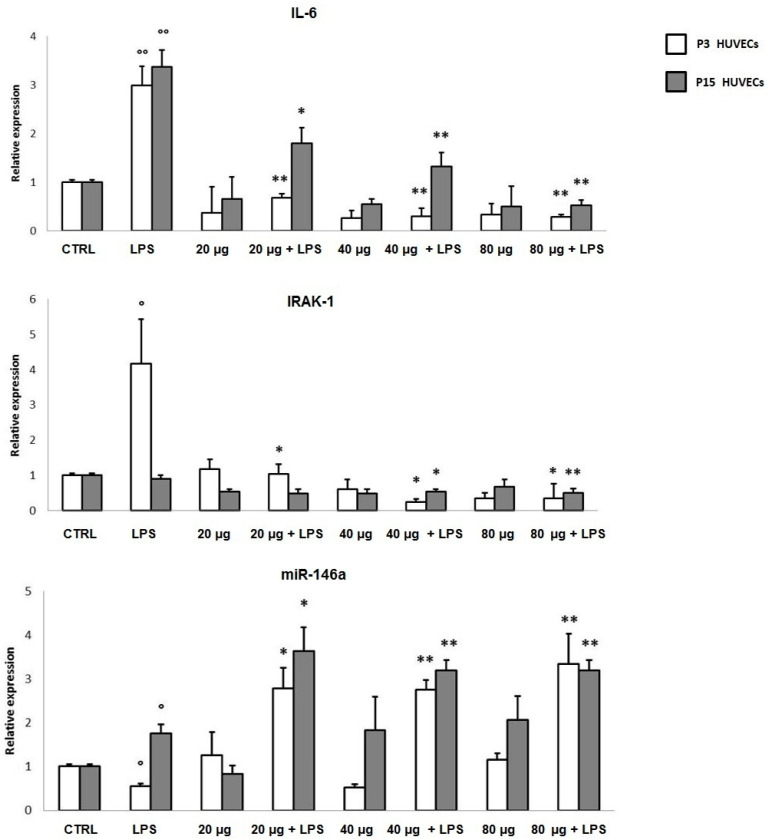
Anti-inflammatory effect of *P. spinosa* extract on IL-6, IRAK-1, and miR-146a expression levels in LPS (lipopolysaccharide, 1 µg/mL) stimulated young (P3) and older (P15) human umbilical vein endothelial cells (HUVECs) after 6 h of treatment. LPS stimulus downregulated miR-146a (LPS). When treated with *P. spinosa* extract during LPS stimulation (20/40/80 µg/mL *P. spinosa* + LPS), cells showed miR-146a upregulation with a decreased expression of both IRAK-1 and IL-6. Results are reported as fold change related to CTRL. Two-tailed paired Student’s *t*-test: * *p* < 0.05 LPS vs. *P. spinosa* + LPS; ** *p* < 0.01 LPS vs. *P. spinosa* + LPS; ° *p* < 0.05 LPS vs. CTRL; °° *p* < 0.05 LPS vs. CTRL.

**Figure 3 antioxidants-10-00374-f003:**
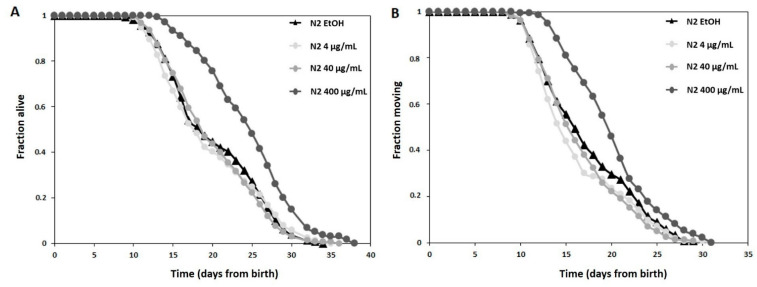
High concentrations (400 µg/mL) of *Prunus* extract extend lifespan and healthspan in *C. elegans*. (**A**) Kaplan–Meier survival curves of *C. elegans* treated with the indicated concentrations of *Prunus* extract showing extended lifespan at the highest concentration. (**B**) Kaplan–Meier survival curves of *C. elegans* treated with the indicated concentrations of *Prunus* extract showing extended healthspan at the highest concentration. For 3A and B panels, pooled data of 180 worms/condition in three independent replicates are shown. Bonferroni *p*-value ETOH vs. 400 µg/mL 1.2 × 10^−7^ in panel A. Bonferroni *p*-value ETOH vs. 400 µg/mL 0.0007 in panel B. Statistical test: Log Rank test.

**Figure 4 antioxidants-10-00374-f004:**
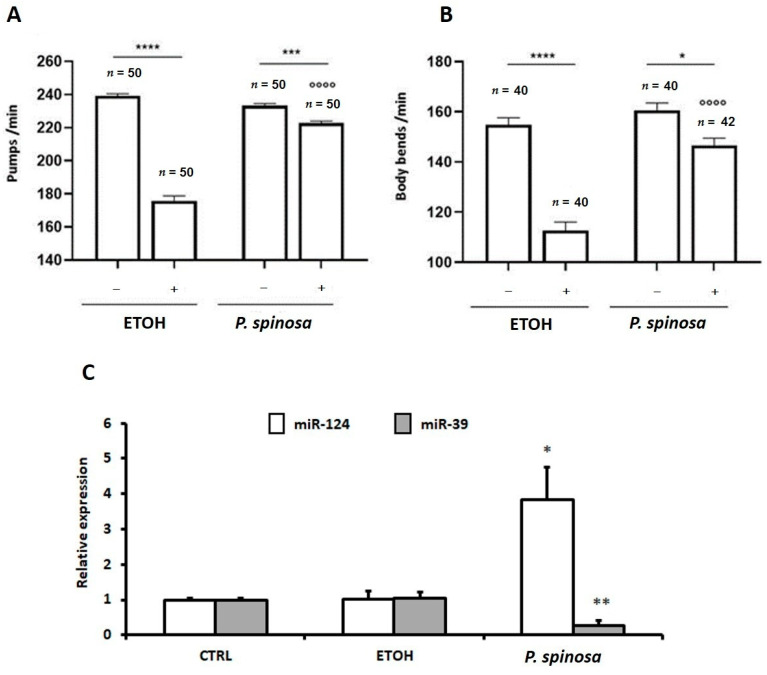
High concentrations of *Prunus* extract protect from H_2_O_2_-induced oxidative stress in *C. elegans*. (**A**) The pharyngeal pumping rate has been measured by counting the number of times the terminal bulb of the pharynx contracted over a 1-min interval (pump/min). Data are shown as mean pumps/min ± SE (*n* = 50 worms/assay, 3 assays). *** *p* < 0.001 and **** *p* < 0.0001, according to two-way ANOVA and Bonferroni’s post hoc test. °°°° *p* < 0.0001 vs. H_2_O_2_-fed worms and grown on ETOH plates, according to two-way ANOVA and Bonferroni’s post hoc test. Interaction < 0.0001. (**B**) The locomotor activity has been evaluated by measuring the number of body bends in liquid over a 1-min interval (body bends/min). Data are shown as mean pumps/min ± SE (*n* = 40 worms/assay, 3 assays). * *p* < 0.05, *****p* < 0.0001, according to two-way ANOVA and Bonferroni’s post hoc test. °°°° *p* < 0.0001 vs. H_2_O_2_-fed worms and grown on ETOH plates, according to two-way ANOVA and Bonferroni’s post hoc test. Interaction < 0.0001. (**C**) MicroRNAs (miR-124 and miR-39) expression levels in *P. spinosa* treated *C. elegans* worms. Treatment of *C. elegans* from embryos with 400 µg/mL of PSF lead to miR-124 upregulation and miR-39 downregulation in the adults. Two-tailed paired Student’s *t*-test: * *p* < 0.05 vs. CTRL; ** *p* < 0.01 vs. CTRL.

**Figure 5 antioxidants-10-00374-f005:**
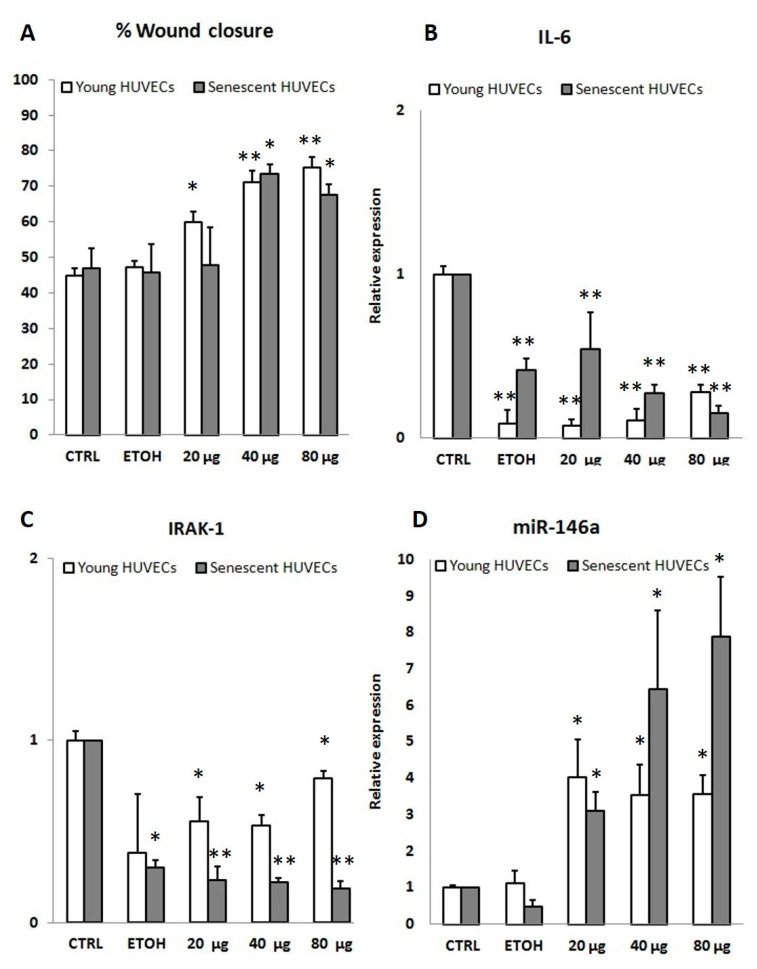
Wound healing (in % wound closure) in *P. spinosa* fruit extract (PSF) treated cells and expression levels of inflammation markers assessed in the same cells. Experimental conditions were the same as those employed for anti-inflammaging assays. (**A**) Percentage of wound healing closure related to CTRL in young (P3) and senescent (P15) HUVECs after 48 h of *P. spinosa* treatment. Our data showed, both in young and older cells, a significant improvement (up to 70%) of wound healing closure (expressed in percentage related to untreated cells) through cell migration. The same cells were detached and used for the evaluation of expression levels of anti-inflammatory markers, IL-6 (**B**), IRAK-1 (**C**), and miR-146a (**D**). MiR-146a upregulation was observed during PSF extract treatment (20/40/80 µg/mL *P. spinosa*). PSF extract treatment caused a decreased expression of both IRAK-1 and IL-6, thus revealing a downregulation of the inflammatory response. Results are reported as fold change related to CTRL. Two-tailed paired Student’s *t*-test: **p* < 0.05 vs. CTRL; ***p* < 0.01 vs. CTRL.

**Figure 6 antioxidants-10-00374-f006:**
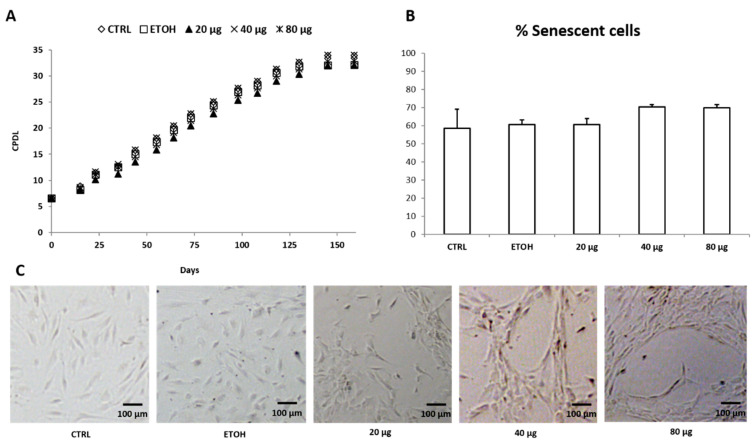
Senescence of HUVECs treated with *P. spinosa* at different concentrations (20, 40, and 80 µg/mL) during 160 days (P2-P15). (**A**) Growth curve showing the cumulative population doubling level (CPDL) during *P. spinosa* treatment (P2-P15). (**B**) Percentage of senescent HUVECs (P15) as a fraction of cells expressing SA-β-Gal. (**C**), Cellular morphology of P15 senescent HUVECs after 160-day *P. spinosa* treatment (Optical microscopy, 10x). The samples CTRL, ETOH and 20 µg/mL treated cells showed large cobblestone shaped cells, while cells treated with 40 µg/mL and 80 µg/mL *P. spinosa* extract showed long cells (spindle-shaped) with multiple protrusions (connecting neighboring cell clusters).

**Figure 7 antioxidants-10-00374-f007:**
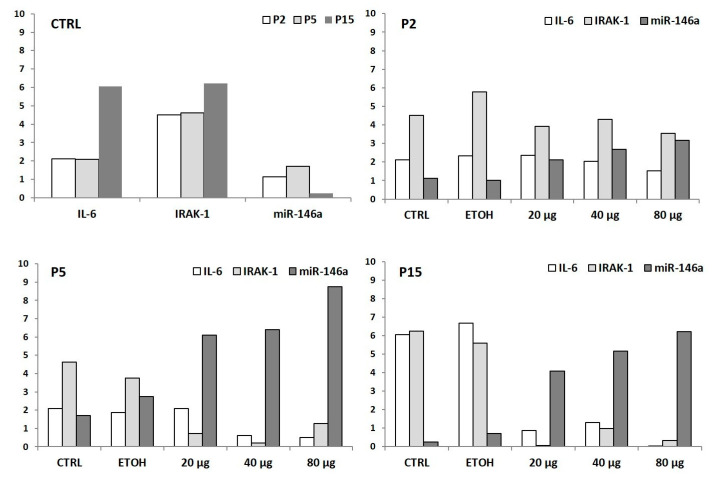
Anti-inflammatory effect of *P. spinosa* during HUVEC senescence (observed at P2, P5, and P15). Untreated (i.e., without any additional triggers) CTRL old cells showed increased IL-6/IRAK-1 expressions and miR-146a downregulation. Notably, in older cells, *P. spinosa* treatments (20/40/80 µg/mL) decreased IL-6/IRAK-1 expressions and upregulated miR-146a, showing a modulation of miR-146a expression level and IL-6 and IRAK-1 amounts comparable to what was observed in younger cells. The relative expression has been determined using the 2^−ΔCt^ method and is expressed as fold change related to the TATA binding protein (TBP) endogenous control.

**Figure 8 antioxidants-10-00374-f008:**
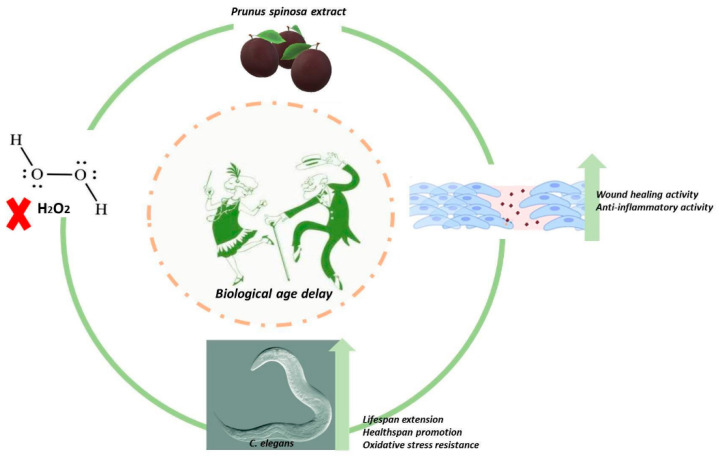
PSF extract delays biological age by acting as an antioxidant and anti-inflammatory agent. Beneficial effects promote in vitro endothelial cell functions (wound healing) and in vivo lifespan/healthspan of *C. elegans*.

## Data Availability

Not applicable.

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
