# Peer review of "Antioxidant and Anti-Inflammaging Ability of Prune (Prunus Spinosa L.) Extract Result in Improved Wound Healing Efficacy"

_antioxidants, 2021, doi:10.3390/antiox10030374_

Round 1
Reviewer 1 Report
The manuscript entitled “Antioxidant and anti-inflammaging ability of prune (Prunus spinosa L.) extract result in improved wound healing efficacy”, authored by Colomba and colleagues, deals with the investigation of antioxidant and anti-inflammaging potential of extracts from Prunus spinose. The work is well organized, however some little changes are needed before consider the manuscript suitable for the publication:
The abstract should be completely rewritten. In particular, it is missing of crucial information. For example, authors should provide the background of the current state of art, together to a little part related to the employed methods. Finally, the results should be summarized and the main conclusion should be added. The authors can check the instruction for its preparation in the section of the official page of Antioxidants journal (https://www.mdpi.com/journal/antioxidants/instructions).
Keywords should be words not contained within the title of the manuscript. Since many of these words are present in the title, you strongly advise to modify and replace them.
In the introduction section, authors should add information regarding the use of plants and/or natural phytochemicals extracted from plants that displayed antinflammatory activity. Moreover, the authors should identify and describe Prunus spinose among the plant family and genus.
Concerning the putative identification, authors do not provide any useful information regarding the employed method. In particular, my principal concern is that the identification of punctual compounds was not carried using an HR-MS methodology, but a simply MS/MS. In this case, authors should provide comparison with pure analytical standard. Moreover, the references 13 is not linked to Phenol Explorer Database (that I think that is the original database used from authors to obtain the data).
In subsection 2.7., the authors should report the primers used in their work as table.
Antioxidant activity should be measured using more than one assay to ensure the validation of the obtained results (FRAP, ORAC, ABTS, CuPRAC, etc ...). These are simple, low-cost, time-saving essays that are performed in almost all scientific articles in order to better understand the mechanism of action and the real antioxidant potential of an extract.
Author Response
Reviewer 1:
The manuscript entitled “Antioxidant and anti-inflammaging ability of prune (Prunus spinosa L.) extract result in improved wound healing efficacy”, authored by Colomba and colleagues, deals with the investigation of antioxidant and anti-inflammaging potential of extracts from Prunus spinose. The work is well organized, however some little changes are needed before consider the manuscript suitable for the publication:
The abstract should be completely rewritten. In particular, it is missing of crucial information. For example, authors should provide the background of the current state of art, together to a little part related to the employed methods. Finally, the results should be summarized and the main conclusion should be added. The authors can check the instruction for its preparation in the section of the official page of Antioxidants journal (https://www.mdpi.com/journal/antioxidants/instructions).
Abstract has been rewritten considering the Journal instructions (without headings). In particular, background has been reported in lines 14-17; employed methods: lines 17-20; results: lines 20-26; main conclusion: lines 26-28.
Keywords should be words not contained within the title of the manuscript. Since many of these words are present in the title, you strongly advise to modify and replace them.
We followed the suggestion.
In the introduction section, authors should add information regarding the use of plants and/or natural phytochemicals extracted from plants that displayed antinflammatory activity. Moreover, the authors should identify and describe Prunus spinose among the plant family and genus.
We added the suggested information (lines: 57-58).
Concerning the putative identification, authors do not provide any useful information regarding the employed method. In particular, my principal concern is that the identification of punctual compounds was not carried using an HR-MS methodology, but a simply MS/MS. In this case, authors should provide comparison with pure analytical standard. Moreover, the references 13 is not linked to Phenol Explorer Database (that I think that is the original database used from authors to obtain the data).
Qualitative extract analysis was performed with “tentative characterization” without using standards as already published by other authors in the same Antioxidants journal. Please see this example: https://www.mdpi.com/2076-3921/9/2/166.
We added more information on the employed method adding the link to “Phenol Explorer Database”. Reference 13 is related to our previous reported analysis on Prunus spinosa as described in the manuscript.
In subsection 2.7., the authors should report the primers used in their work as table.
We agree with the Reviewer, but since a lot of tables and figures have been asked to be grouped from previous reviewers, we decided to insert the primers, as other papers do (even published in Antioxidants journal), in the text.
Antioxidant activity should be measured using more than one assay to ensure the validation of the obtained results (FRAP, ORAC, ABTS, CuPRAC, etc ...). These are simple, low-cost, time-saving essays that are performed in almost all scientific articles in order to better understand the mechanism of action and the real antioxidant potential of an extract.
We agree with the Reviewer. For this reason, after validating in vitro the antioxidant capacity of prunus spinosa fruit extract (by DPPH assay), we used an in vivo analysis for the evaluation of the antioxidant properties of PSF. This is a more innovative and sophisticated way to evaluate antioxidant properties after digestion. This is quite important for extracts originating from nutritional plants.
Reviewer 2 Report
The authors have modified the resubmitted manuscript in a suitable way, so it is now more complete and convincing. I have no further comments.
Author Response
Reviewer 2:
The authors have modified the resubmitted manuscript in a suitable way, so it is now more complete and convincing. I have no further comments.
Thank you for appreciating all the revisions performed.
This manuscript is a resubmission of an earlier submission. The following is a list of the peer review reports and author responses from that submission.
Round 1
Reviewer 1 Report
The manuscript by Coppari et al. presents the interesting findings on the beneficial anti-inflammatory effects of Prunus spinosa L. fruit (PSF) ethanol extract against aging-related phenotypes using HUVECs and nematode C. elegans models. Overall, the results are interesting. However, several improvements are needed to be make before acceptance.
Major Comments and Recommendations:
Lane 2: The title of the manuscript has to be more specific.
Lane 109: Should read „The supernatants were...“.
Lane 112: The total amount of dried PSF extract in micrograms extracted from 50 g of fruit and peel should be specified.
Methods section: The extract preparation and DPPH radical assay are the same as published previously by Sabatini et al., J. Funct. Foods, 2020 (Reference 11). Authors should reformulate these sections.
Lanes 271-284: Presented results of the DPPH radical assay are out of range to calculate the EC50. Was the EC50 value approximated? In fact, the tested concentrations of the PSF extract inhibit the DPPH from 2% to 6%. This is in contrast with previously published work where “the same concentration” of 20 micrograms/ml inhibited DPPH close to 25%. This discrepancy should be explained, otherwise the whole concept of the beneficial antioxidant properties of PSF extract tested in this study and presented in the manuscript is questionable.
Figure 2: The X-axis label of IRAK-1 expression is missing.
Lanes 343-344: Should be part of the Legend to the Figure 3.
Lanes 383-384: Should be part of the Legend to the Figure 5.
Results of the cellular senescence at molecular level (including the Figure 9) must be omitted (too speculative) as they are based on a single experiment and no statistical analysis was performed. Discussion should be modified and rewritten accordingly.
Author Response
Authors wish to thank both reviewers for their valuable comments and suggestions which improved the manuscript. Authors’ notes are in red.
REVIEWER #1
The manuscript by Coppari et al. presents the interesting findings on the beneficial anti-inflammatory effects of Prunus spinosa L. fruit (PSF) ethanol extract against aging-related phenotypes using HUVECs and nematode C. elegans models. Overall, the results are interesting. However, several improvements are needed to be make before acceptance.
Major Comments and Recommendations:
Lane 2: The title of the manuscript has to be more specific.
As suggested also by reviewer #2, the title has been modified.
The new title is: “Antioxidant and anti-inflammaging ability of prune (Prunus spinosa L.) extract result in improved wound healing efficacy”.
Please note that the new title represents the major conclusion of the study because, as reported in the revised version of the ms (manuscript), in this study our attention focused on a further characterization of PSF ethanol extract in order to evaluate its: i) antioxidant, anti-inflammatory and potential anti-aging properties; and ii) wound healing efficacy. After assessing PSF antioxidant and anti-inflammaging ability both in vitro and in vivo, we evaluated P. spinosa potential on wound healing which was the main purpose of this study. In fact, our work hypothesis was that prunus extract could exert a positive effect on wound healing as its bioactive polyphenols exhibit antioxidant, anti-inflammatory and anti-aging properties,which makes the extract ideal to play such a role in tissue repair.
Lane 109: Should read „The supernatants were...“.
Lane 112: The total amount of dried PSF extract in micrograms extracted from 50 g of fruit and peel should be specified.
Both points have been modified and the data required has been specified (the total amount of dried PSF extract extracted from 50 g of fruit and peel is 11.6 g, see ms).
Methods section: The extract preparation and DPPH radical assay are the same as published previously by Sabatini et al., J. Funct. Foods, 2020 (Reference 11). Authors should reformulate these sections.
Lanes 271-284: Presented results of the DPPH radical assay are out of range to calculate the EC50. Was the EC50 value approximated?
In fact, the tested concentrations of the PSF extract inhibit the DPPH from 2% to 6%. This is in contrast with previously published work where “the same concentration” of 20 micrograms/ml inhibited DPPH close to 25%. This discrepancy should be explained, otherwise the whole concept of the beneficial antioxidant properties of PSF extract tested in this study and presented in the manuscript is questionable.
Parts dealing with the extract preparation and DPPH radical assay have been corrected and partially rewritten. As for the extract preparation, since the method is exactly that reported in Sabatini et al., J. Funct. Foods, 2020 (Reference 11), rather than reformulating the sentences using synonyms, we preferred to refer directly to the paper, reporting only the different composition of the homogenizing solution (which was an acidified aqueous ethanol homogenizing solution).
We corrected the linear regression of DPPH and recalculated EC50. In particular, the equation obtained by linear regression analysis was used to calculate the EC50 (considering y=50). New EC50 = 64.2 µg/ml [calculated (given y=50) as: 50 = 0.502x+17.906, see legend of figure 1].
Please note that discrepancy with data previously published (present study EC 50 = 64.2 µg/ml and EC50 by Sabatini et al. 2020 = about 300 µg/ml) is due to the fact that the extract used by Sabatini et al., had a slightly different chemical composition (in this work the extraction has been improved, which has now been specified and clarified in the revised version) and a different antioxidant activity (EC50 calculated in the present study is more than 4 times higher). Finally, note that once the math has been corrected, the EC50 (64.2 µg/ml) now falls within the range of PSF concentrations used in our experimental conditions (20, 40, 80 µg/ml).
Figure 2: The X-axis label of IRAK-1 expression is missing.
The X-axis label of figure 2 has been added.
Lanes 343-344: Should be part of the Legend to the Figure 3.
Lanes 383-384: Should be part of the Legend to the Figure 5.
As suggested by the reviewer #2, the figures have been combined to reduce their number, and the legends modified accordingly.
Results of the cellular senescence at molecular level (including the Figure 9) must be omitted (too speculative) as they are based on a single experiment and no statistical analysis was performed. Discussion should be modified and rewritten accordingly.
Given the complexity of the work and the time (the experiment lasts 160 days) it takes to carry out this type of experiment in triplicates using cells (HUVECs) which, moreover, hardly replicate as they age, we believe that this result, despite being preliminary, may be interesting per se and also interesting for other colleagues who work in this field. In fact, considering that the data on cellular senescence, for the reasons mentioned above, are very few, we hope our data could still be considered a starting point for subsequent experiments.

Reviewer 2 Report
In this manuscript, Prunus Spinosa extracts (PSE) has been demonstrated to have beneficial effects on oxidative damage, senescence, inflammation, and wound healing in cellular systems. These effects have been investigated using many different experimental approaches using HUVEC cells as well as C. elegans. On the other hand, experiments are so diverse that it is difficult to realize the logical connections between them. It would be necessary to make more clear what are the main topic and major conclusion of this paper. Specific comments are as follows.
1, The title should be what represents the major conclusion of this work. The current title says "SPE improves would healing", but wound healing effect was shown only in Fig. 6. In addition, "biological aging" is a little unclear, and might be better to change to a more direct expression like "upregulation of miR-146a, a senescence-sensitive factor" if it is a new finding. It is also difficult for me to resolve the grammatical structure of the title sentence.
2, In general, it is not recommended to use different experimental systems such as HUVEC and C. elegans in a single report. In this paper, it is more confusing because data from HUVEC and C. elegans are presented in a mixed manner (HUVEC:Fig.2,6,7,8,9, C.elegans:Fig.3,4,5). It would be best to prepare two separate papers. However, if they are reported together in a single paper, it is better to reduce the number of figures regarding C. elegans to make clear which is the major topic. For example, it might be possible to combine Fig. 3, 4 and 5 to a single figure and move to the last part of this paper to discuss the similarity with HUVEC.
3, Fig. 2. shows that PFS upregulated miR-146a in P15 HUVEC especially in the presence of LPS, while it was described in the text line 294 that the data "confirm ...." . If it is a new finding, it should be described like "it was found that ...". If it is a confirmation of a previous report, the paper should be quoted.
4, The graph in Fig. 6 should be taller (i.e. should be depicted with a larger y-axis/x-axis ratio) so that the differences between bars could be seen more clearly.
5, In Fig. 7, IL6 and IRAK-1 were shown to be downregulated not only by PSE but also by EtOH alone, indicating that these effects could be due to EtOH, the vehicle. This is in contrast to miR-146a, which was upregulated by PSE but not by EtOH alone. It is necessary to explain the data, or omit the chart about IL6 and IRAK-1.
6, Student t-test could be applied only when only two groups are present in the data set. If two out of three or more groups are to be compared, ANOVA and a post-hock test are required.
Author Response
Authors wish to thank both reviewers for their valuable comments and suggestions which improved the manuscript. Authors’ notes are in red.
REVIEWER #2
In this manuscript, Prunus Spinosa extracts (PSE) has been demonstrated to have beneficial effects on oxidative damage, senescence, inflammation, and wound healing in cellular systems. These effects have been investigated using many different experimental approaches using HUVEC cells as well as C. elegans. On the other hand, experiments are so diverse that it is difficult to realize the logical connections between them. It would be necessary to make more clear what are the main topic and major conclusion of this paper. Specific comments are as follows.
1, The title should be what represents the major conclusion of this work. The current title says "SPE improves would healing", but wound healing effect was shown only in Fig. 6. In addition, "biological aging" is a little unclear, and might be better to change to a more direct expression like "upregulation of miR-146a, a senescence-sensitive factor" if it is a new finding. It is also difficult for me to resolve the grammatical structure of the title sentence.
We changed the title in: “Antioxidant and anti-inflammaging ability of prune (Prunus spinosa L.) extract result in improved wound healing efficacy”.
Please note that the new title represents the major conclusion of the study because, as reported in the revised version of the ms, in this study our attention focused on a further characterization of PSF ethanol extract in order to evaluate its: i) antioxidant, anti-inflammatory, and potential anti-aging properties; and ii) wound healing efficacy. After assessing PSF antioxidant and anti-inflammaging abilities both in vitro and in vivo, we evaluated P. spinosa potential on wound healing which was the main purpose of this study. Please note that the PSF wound healing effects are not only shown in figure 6 (wound closure) but inflammaging markers have been analyzed in young and senescent HUVECs after wound healing and are showed in Fig. 7 (in the revised manuscript figure 6 and 7 have been combined in a unique figure, see revised figure 5).
The main and original conclusion of the study is that prunus extract exerts a positive effect on wound healing in HUVEC cells, as its bioactive polyphenols exhibit antioxidant and anti-inflammatory properties, which makes the extract an ideal candidate to play such a role in tissue repair in a proper cellular model (HUVECs).
2, In general, it is not recommended to use different experimental systems such as HUVEC and C. elegans in a single report. In this paper, it is more confusing because data from HUVEC and C. elegans are presented in a mixed manner (HUVEC:Fig.2,6,7,8,9, C.elegans:Fig.3,4,5). It would be best to prepare two separate papers. However, if they are reported together in a single paper, it is better to reduce the number of figures regarding C. elegans to make clear which is the major topic. For example, it might be possible to combine Fig. 3, 4 and 5 to a single figure and move to the last part of this paper to discuss the similarity with HUVEC.
In the revised text we have tried to better clarify the organization of the project separating the in vitro and in vivo experimental parts and, accordingly, combining some figures.
To this aim, after assessing the chemical composition of the extract, the study was organized in two experimental sections, one in vitro and the other one in vivo. The antioxidant activity was tested by DPPH and after that, the anti-inflammatory effect was tested on young (P3) and senescent (P15) HUVEC cells, stimulated with LPS, by measuring miR-146a, along with its cell IRAK-1 target and IL-6 expression levels.
After establishing in vivo antioxidant and in vitro anti-inflammaging PSF ability, we evaluated P. spinosa potential on wound healing which was the main purpose of this study. In fact, our work hypothesis was that prunus extract could exert a positive effect on wound healing as its bioactive polyphenols exhibited antioxidant, anti-inflammatory, and anti-aging properties, making the extract ideal to play such a role in tissue repair (a complex molecular and cellular process made of several overlapping phases including inflammation).
As far as concerns the relevance of in vivo analyses, generally, when examining living organisms, the bioavailability has a key role while evaluating health effects. That is the reason why concentrations employed in in vivo assays were different and higher that those tested in vitro (see results section 3.3.1).
Finally, considering the reviewer’s statement that “it is not recommended to use different experimental systems”, please note that there are plenty of studies in the literature, which use different experimental systems to support the evolutionary conserved effects of genetic or extrinsic (diet, drugs, or environmental factors) interventions and to increase the power of the data through in vitro and in vivo experimental approaches, thus ultimately strengthening the findings.
Just to mention some examples specifically coming from the authors:
- The flavonoid 4,4'-dimethoxychalcone promotes autophagy-dependent longevity across species. Nat Commun. 2019 Feb 19;10(1):651. doi: 10.1038/s41467-019-08555-w.
- Iron-Starvation-Induced Mitophagy Mediates Lifespan Extension upon Mitochondrial Stress in C. elegans. Curr Biol. 2015 Jul 20;25(14):1810-22. doi: 10.1016/j.cub.2015.05.059. Epub 2015 Jul 2.
- The aryl hydrocarbon receptor promotes aging phenotypes across species. Sci Rep. 2016 Jan 21;6:19618. doi: 10.1038/srep19618.
- -HDAC inhibition improves autophagic and lysosomal function to prevent loss of subcutaneous fat in a mouse model of Cockayne syndrome. Sci Transl Med. 2018 Aug 29;10(456):eaam7510. doi: 10.1126/scitranslmed.aam7510.
3, Fig. 2. shows that PFS upregulated miR-146a in P15 HUVEC especially in the presence of LPS, while it was described in the text line 294 that the data "confirm ...." . If it is a new finding, it should be described like "it was found that ...". If it is a confirmation of a previous report, the paper should be quoted.
The text has been rewritten clarifying that we refer to a better quality extract and a different cell system (HUVEC vs U937, under the same experimental conditions). Please note that in Sabatini et al., 2020 (Ref. 11), the anti-inflammatory activity has been partially explored by using a similar prunus extract but a different cell type (i.e. non endothelial U937 monocytes) and the main obtained results were focused on the prunus extract antimicrobial ability. Instead, in this ms, the results are in line with those previously described considering only the anti-inflammatory activity, but are, at the same time, data on the antioxidant and anti-inflammaging ability of the prunus extract are original since they were obtained in a cell system (HUVEC) more suitable for the purpose of our study: the evaluation of prunus extract effect on wound healing.
4, The graph in Fig. 6 should be taller (i.e. should be depicted with a larger y-axis/x-axis ratio) so that the differences between bars could be seen more clearly.
The graph has been arranged as suggested.
5, In Fig. 7, IL6 and IRAK-1 were shown to be downregulated not only by PSE but also by EtOH alone, indicating that these effects could be due to EtOH, the vehicle. This is in contrast to miR-146a, which was upregulated by PSE but not by EtOH alone. It is necessary to explain the data, or omit the chart about IL6 and IRAK-1.
Results observed in IL-6 and IRAK-1 expression levels may appear, in part, unexpected. In fact, as shown in the results section (see revised Fig. 5B-C), IL-6 and IRAK-1 were down-regulated not only by PSF extract but also by EtOH alone (the vehicle). This result, although surprising, was constant and repetitive (experiments conducted in triplicate) and, since the very beginning, quite difficult to interpret. In fact, given the pro-inflammatory effect of EtOH, one would have expected exactly the opposite (i.e. high levels of IL-6 and IRAK-1). However, considering that this apparent "anomaly" occurred only in PCR reactions performed on (messenger or micro) RNA extracted from HUVEC cells, a possible explanation is that it is a technical bias. In fact, taking into account that ethanol is also a PCR inhibitor (in the case of real time PCR, it acts as a reverse transcriptase inhibitor), we hypothesize that a very small percentage of unintentional EtOH contamination during the ribonucleic acid extraction process may have inhibited the amplification reaction. This would be indirectly confirmed by the fact that no anomalies were noted in the experiments carried out directly on whole cells. In addition, please note that all the experiments were conducted with non-toxic EtOH concentration levels and that the cells were always absolutely vital and did not present any type of stress. Finally, the fact that molecular amplifications on C. elegans did not show any bias could be due to the fact that in vivo the penetration of EtOH into the living tissues is considerably lower than that which occurs in an in vitro system, thus reducing the risk of PCR inhibition.
6, Student t-test could be applied only when only two groups are present in the data set. If two out of three or more groups are to be compared, ANOVA and a post-hock test are required.
Comparisons were always performed considering two experimental conditions at a time (i.e. pairwise comparisons). For this reason we believe that Student’s t-test was properly applied; moreover, in figure legends the compared groups are always indicated. While, in in vivo experiments, ANOVA and a post-hoc test were applied to simultaneously compare different experimental conditions.

Round 2
Reviewer 1 Report
I thank the authors for the attention given to my comments. They have responded satisfactorily to my concerns. I have no further suggestions.
Reviewer 2 Report
1.
> We changed the title in: “Antioxidant and anti-inflammaging ability
> of prune (Prunus spinosa L.) extract result in improved wound
> healing efficacy”.
Now I have realized that this paper intended to demonstrate wound healing effect of PSF. However, wound healing effect was examined only in the experiment shown in Fig. 5a. On the other hand, the rest of figures are related to the antioxidant activity, anti-inflammatory activity and senescence, which are not directly related to wound healing. Although the new title claims that the anti-oxidant and anti-inflammaging activity "results in" improved wound healing, no evidence was presented to demonstrate the direct relevance between them.
In other words, the conclusion of this paper was obtained only from Fig.5a. In my opinion, Fig. 5a is not enough to construct a complete paper to be published.